# Ιn Vitro Screening of γ-Aminobutyric Acid and Autoinducer-2 Signalling in Lactic Acid Bacteria Exhibiting Probiotic Potential Isolated from Natural Black Conservolea Olives

**DOI:** 10.3390/foods8120640

**Published:** 2019-12-04

**Authors:** Foteini Pavli, Eleni Gkana, Oluwabunmi Adebambo, Kimon-Andreas Karatzas, Efstathios Panagou, George-John E. Nychas

**Affiliations:** 1Laboratory of Microbiology Biotechnology of Foods, Department of Food Science and Human Nutrition, School of Food, Biotechnology and Development, Agricultural University of Athens, 11855 Athens, Greece; photpavli@gmail.com (F.P.); ganaelina@gmail.com (E.G.); stathispanagou@aua.gr (E.P.); 2Department of Food and Nutritional Sciences, University of Reading, Reading RG6 6AP, UK; oluade@reading.ac.uk (O.A.); k.karatzas@reading.ac.uk (K.-A.K.)

**Keywords:** probiotic potential, lactic acid bacteria, γ-aminobutyric acid, autoinducer-2, table olives

## Abstract

In the present study, 33 strains of lactic acid bacteria (LAB) previously isolated from natural black Conservolea olives were assessed for their probiotic potential in vitro, as well as for their autoinducer-2 (AI-2) activity under standard growth conditions and the production of γ-aminobutyric acid (GABA). The probiotic tests included the in vitro resistance to low pH and resistance to bile salts, the evaluation of bile salt hydrolase activity, as well as safety tests regarding their possible haemolytic activity and their antimicrobial activity against pathogens. The results indicated that 17 strains were able to survive in low pH and in the presence of bile, with 15 of them also exhibiting partial bile salt hydrolase activity. None of the strains exhibited haemolytic activity or inhibited the growth of any of the examined pathogens. Moreover, the strains displayed generally low AI-2 activity under the growth conditions tested, regardless of the species. Interestingly, in contrast to what has been found in most foods, none of the isolates were found to produce GABA after 48 h of growth. The results from the AI-2 activity and extracellular GABA detection were considered as unexpected for LAB with probiotic attributes.

## 1. Introduction

Lactic acid bacteria (LAB) are the most important group of probiotic bacteria, followed by *Bifidobacteria*, and some species of *Enterococci* and *Bacillus*, although the safety of the latter remains a controversy. Many beneficial effects of probiotics have been associated with lowering of cholesterol, controlling symptoms of lactose intolerance, stimulating the immune system, relieving constipation, preventing antibiotic-associated and traveller’s diarrhoea, and prevention of *Clostridium difficile* infections [1,2,3,4,5]. Such bacteria have been isolated from various sources, including fermented dairy products [6,7], non-dairy products [8,9,10,11,12], and from the intestinal tract of healthy individuals or infants [13,14,15]. Fermented products of plant origin, such as table olives, are considered an excellent source of probiotic bacteria and also great vehicles for probiotic delivery, with many studies confirming their potential [16,17,18,19]. A series of in vitro tests could be performed as a first step in order to assess the probiotic potential of these strains. Such tests could include acid and bile resistance, the production of antimicrobial compounds, and bile salt hydrolase activity, although further tests and clinical studies are also required at a later stage.

Cell-to-cell communication, also called quorum sensing, is the process of signalling that enables bacteria to control their gene expression and regulate their activities. A “universal” signal-molecule called autoinducer-2 (AI-2) could be present in gram-positive and gram-negative bacteria, supporting inter-species communication. The expression of various phenotypes including virulence factors, biofilm formation, light production, and stress resistance are only some of the functions that are linked to AI-2 in many bacterial species [20,21]. This is of a particular interest in the case of probiotic bacteria, due to their exposure to a variety of environmental challenges, including low pH, the presence of bile and enzymes. Lebeer [22] reported that AI-2 production had an impact on several physiological functions of the probiotic *Lactobacillus rhamnosus* GG. In another study, *Lactobacillus paraplantarum* L-ZS9 was found to overexpress *luxS* gene and produce AI-2 that enhances heat, bile, and salt resistance, while also having an effect on its biofilm formation [23]. Furthermore, other studies have related AI-2 activity with acid tolerance in probiotic bacteria [24].

γ-Aminobutyric acid (GABA) is an important metabolite produced by the glutamate decarboxylase system (GAD) in various bacteria through the decarboxylation of glutamate under acidic conditions. Thus, quantification of GABA could be an important tool to assess acid resistance of some probiotic bacteria, which is important because survival in the acidic pH of the stomach is required for gut colonization in the host [25]. Antianxiety, antidepressant, antihypertensive, and tranquilizing roles have been attributed to GABA, thus far [26,27]. Due to the aforementioned beneficial effects, much attention has been paid to the development of foods enriched with GABA. Several microorganisms have been examined for their potential to produce GABA through the GAD system, such as *Lactobacillus brevis* and *Lactococcus lactis*. Furthermore, it has been reported that strains of *Lactobacillus buchneri*, *Lactobacillus paracasei*, and *Lactobacillus plantarum* isolated from traditional cheeses have the ability to produce GABA [28,29]. It has to be noted that the ability of LAB to produce GABA varies among species and strains [30]. Therefore, screening of LAB for their ability to synthesize GABA is of special interest for the development of novel GABA-enriched functional foods, and this ability of LAB might be considered as an important probiotic trait in the near future [31].

In the present study, 33 strains of LAB were examined for their probiotic potential using a series of in vitro tests. The strains that exhibited good performance in the probiotic tests were then screened for AI-2 activity under standard growth conditions, followed by another screening with regard to their ability to produce GABA.

## 2. Materials and Methods

A total of 33 strains of LAB, previously isolated from natural black Conservolea olives during storage in different packaging conditions, as well as two reference strains, namely *Lactobacillus casei* Shirota (ACA-DC 6002) and *Lactobacillus rhamnosus* GG (ATCC 53103), kindly provided by Prof. E. Tsakalidou (Laboratory of Dairy Science, Agricultural University of Athens) were screened for their probiotic potential following a series of in vitro tests. The studied strains were 8 *Lactobacillus plantarum*, 20 *Lactobacillus pentosus*, 1 *Lactobacillus paraplantarum*, 2 *Lactobacillus corinyformis*, and 2 *Pediococcus ethanolidurans* (Table 1), that were previously isolated, identified, and characterized [32]. Strains were stored in 20% glycerol at −80 °C for long-term storage, and were routinely cultured at 30 °C in de Man, Rogosa, and Sharpe (MRS) broth (Biolife, Milan, Italy) or agar (Biolife, Italy) for 24 or 18 h and 72 h, respectively.

### 2.1. Low pH Assay, Bile Salts Assay, and Bile Salt Hydrolase (BSH) Activity

The tests to assess the resistance of the strains to low pH and bile salts, as well as the bile salt hydrolase (BSH) activity, were performed according to Argyri et al. [10]. Briefly, for the resistance to low pH, bacterial cells from overnight cultures (18 h) were harvested by centrifugation (5000× *g*, 15 min, 4 °C), washed twice with phosphate-buffered saline (PBS) (pH 7.2), and finally re-suspended in PBS solution with a pH of 2.5. After incubation for 0, 1, 2, and 3 h at 37 °C under stirring conditions, resistance to low pH was assessed in terms of viable colony counts enumerated on MRS agar. The strains that exhibited final counts ≥6 log CFU/mL at a pH of 2.5 for 3 h were selected to be screened for bile salt resistance. For the bile salt resistance test, the same procedure was applied, with the final resuspension being in PBS solution with a pH 8, containing 0.5% bile salts (Oxoid, Hampshire, UK). After incubation for 0, 1, 2, 3, and 4 h at 37 °C under stirring conditions, resistance was assessed in terms of viable colony counts enumerated on MRS agar. The strains that exhibited final counts ≥6 log CFU/mL under these conditions were selected to be screened for BSH activity. For the BSH activity, the bacterial cultures were streaked on MRS agar containing 0.5% taurodeoxycholic acid (TDCA; Sigma, St. Louis, MO, USA). The hydrolysis effect was assessed by different colony morphology in comparison to the control MRS plates (without TDCA) after anaerobic incubation at 37 °C for 48 h. The results were expressed as no hydrolysis or partial hydrolysis. The assays of low pH and bile were performed in triplicate, whereas the test for BSH activity was performed in duplicate.

### 2.2. Safety Assessment of the Selected Strains

The selected strains from the previous tests were further evaluated regarding their potential haemolytic activity and antimicrobial activity according to Argyri et al. [10]. Fresh bacterial cultures were streaked on Columbia agar plates (Lab M Limited, Lancashire, United Kingdom) containing 5% *w*/*v* horse blood and incubated for 48 h at 30 °C. After incubation, the plates were examined for signs of α-haemolysis (green-hued zones around colonies), β-haemolysis (clear zones around colonies), or γ-haemolysis (no zones around colonies). With regard to the antimicrobial activity, all strains were tested against the following pathogens: *Listeria monocytogenes* ATCC 13932; *Listeria monocytogenes* FMCC B-129; *Listeria monocytogenes* 23UD, kindly provided by Prof L. Cocolin [33]; *Listeria monocytogenes* H7550, kindly provided by Prof. S. Kathariou; *Salmonella enterica* subsp. *enterica* serovar Enteritidis P167807; *Salmonella enterica* subsp. *enterica* serovar Typhimurium 4/74 [34]; *Salmonella enterica* FMCC B-64; *Salmonella enteritidis* ATCC 13076; *Escherichia coli* FMCC B-13; *Escherichia coli* NCTC 13127; *Escherichia coli* ATCC 35150; and *Escherichia coli* ATCC 25922. The tests for haemolytic and antimicrobial activity were performed in duplicate.

### 2.3. Screening for AI-2 Activity

For this assay, two strains of *Vibrio harveyi* were used: *V. harveyi* BAA-1117 (ATCC BB-170) as biosensor and *V. harveyi* BAA-1119 (ATCC BB152) as positive control. *Vibrio* strains were cultured in Autoinducer Bioassay (AB) broth [35], and incubated with agitation (160 rpm) at 30 °C for 24 h. The cultures used directly in the bioassay were prepared by transferring a single colony from AB agar in 10 mL of AB broth, and incubating with agitation (160 rpm) at 30 °C for 16 h. For the growth of the LAB strains, quarter-strength brain heart infusion (BHI) broth (Lab M Limited, Lancashire, United Kingdom) was used as previously reported [36]. The AI-2 activity bioassay was performed as described previously [37]. In this study, 10 μL of sterile growth medium was used as a negative control, whereas 10 μL of the cell-free supernatant (CFS) from *V. harveyi* BA-1119 strain was used as a positive control to verify the bioassay. The microplates were incubated at 30 °C and luminescence was measured every 15 min using a Synergy HT multi-mode microplate reader (Biotek, Winooski, VT, USA). Measurements were collected until the negative control exhibited an increase in luminescence. AI-2 activity was expressed as relative AI-2 activity, which was calculated as the ratio of luminescence of the test sample (CFS_LAB_) to that of the control (negative) sample. The bioassay was performed in triplicate with four technical replicates each.

### 2.4. Detection of Extracellular GABA

Single colonies from each strain, previously grown on MRS agar, were obtained and inoculated in MRS broth, followed by incubation anaerobically at 37 °C overnight. These cultures were used as the inoculum 1% (*v*/*v*) to prepare the cultures used for the GABA determination in MRS broth supplemented with 10 mM L-glutamic acid (Sigma-Aldrich, Poole, United Kingdom) and were incubated under the same conditions. The population of each strain together with the pH values with the presence of L-glutamic acid after 48 h incubation were recorded, whereas the supernatant was collected from each culture with centrifugation (13,000 rpm for 10 min). Then, 10 μL of the supernatant was incubated with 90 μL of the assay mixture in each well of a 96-well microtiter plate, as described previously [38,39]. Additionally, standard solutions containing 0, 1, 2, 3, 4, 5, 6, 7, 8, 9, and 10 mM GABA were added to 90 μL of the assay mixture on the same microtiter plate, in order to obtain a standard curve for GABA determination. The microtiter plate was incubated at 37 °C for 180 min using a Sunrise Spectrophotometer (Tecan, Männedorf, Switzerland), and the optical density (OD) at 340 nm was measured every 120 s. The concentration of GABA in the supernatant was calculated using the calibration curves generated by the standard solutions. The test for GABA detection was performed in triplicate.

### 2.5. Statistical Analysis

The statistical analysis was performed using SPSS for Windows, Version 16.0 (SPSS Inc., Chicago, IL, USA). Regarding the resistance to low pH and resistance to bile salts, analysis of variance (ANOVA) for final bacterial counts of each strain (per hour and in total) was performed and means were separated with Tukey’s HSD test. The Tukey post hoc test was also used to compare the means of GABA concentration and AI-2 activity, expressed as the ratio of luminescence of the test sample to the control sample. All differences were reported at a significance level of 0.05.

## 3. Results and Discussion

### 3.1. Low pH Assay, Bile Salts Assay, and BSH Activity

Out of the 33 LAB strains, 18 exhibited high population counts after exposure to pH of 2.5 for a total of 3 h. Amongst these, eight strains (five *Lactobacillus pentosus* and three *Lactobacillus plantarum*) showed the highest population ≥7 log CFU/mL under the acidic conditions tested (Figure 1). Statistically significant differences in the bacterial counts were detected after 1 h in low pH (*p* < 0.05). Results from low pH resistance were in agreement with other studies, where *Lactobacillus pentosus* and *Lactobacillus plantarum* strains were able to maintain their viability after exposure to low pH; however, a variation in the acid resistance among different strains was also observed [6,8,10]. Regarding the resistance to bile salts, out of the 18 strains, only 1 (*Pediococcus ethanolidurans* B389) showed final population <6 log CFU/mL after exposure to bile salts for 4 h. A total of 16 out of 17 strains exhibited a very low log reduction <1, whereas one strain, *Lactobacillus pentosus* B362, showed a slightly higher log reduction (1.14 log CFU/mL; Figure 2). The resistance to bile salts is a prerequisite for probiotic characterization, and guarantees that the cells could reach the intestinal tract alive [40]. Although generally lactobacillus species are able to tolerate bile concentrations normally encountered in the host (0.1%–0.5%), great variability in bile resistance has been reported at genus and species level [41,42]. These observations confirm the hypothesis that bile resistance is a strain-dependent characteristic [43]. It has to be noted that bacterial resistance to low pH and bile salts in vitro is not necessarily similar to the in vivo behaviour [44].

BSH was examined on those strains that exhibited satisfactory performance in the low pH and bile resistance tests. A total of 15 out of 17 strains exhibited partial BSH activity, expressed as differentiated colony morphology recorded on TDCA-MRS agar plates compared to the control MRS agar plates. BSH activity might be a beneficial attribute for a potentially probiotic strain and is often associated with the resistance to bile salts due to the reduction in serum cholesterol or cholesterol solubility and absorption [45]. However, further studies are needed to confirm that other risks for the host are kept to a minimum when excessive amount of probiotic bacteria is consumed. Furthermore, apart from live cells, dead or non-growing lactobacilli cells have been reported to have the ability to reduce cholesterol from media [46].

### 3.2. Safety Assessment of the Selected Strains

Haemolytic activity is one of the screening tests performed for probiotic characterization. Absence of haemolytic activity is considered as a safety requirement for the selection of probiotic strains. In the present study, none of the 17 strains, previously selected for their good performance in low pH and bile assays, exhibited α- or β-haemolysis, whereas all of them exhibited γ-haemolysis (no haemolysis) (Table 2). The findings are similar to those of previous studies regarding the safety of the LAB [6,10,47,48,49], although some exceptions do exist.

The antimicrobial activity against pathogens is deemed as a desirable trait, although not obligatory for probiotic characterization. From the 17 selected LAB strains, none of them were found to inhibit the growth of the 12 pathogens tested, according to the well-diffusion method. These results are in agreement with previous studies [6,10,50], where no antimicrobial activity was reported. On the contrary, significant antimicrobial activity of the LAB against two strains of *Listeria monocytogenes* and low or moderate antimicrobial activity against *Bacillus cereus* was reported in another study [48].

### 3.3. AI-2 Activity

The AI-2 activity for the tested strains exhibited values ranging from 0.32- to 1.57-fold compared to the negative control, whereas the bacterial counts were estimated from 6.42 to 8.30 log CFU/mL after 20 h of incubation time in ¼ strength BHI (Table 3). According to the results obtained, the selected strains exhibiting probiotic potential did not show detectable AI-2 activity under these growth conditions. Generally, diverse results have been reported with regard to the ability of LAB to produce AI-2 molecules [24,36,51]. The detection of AI-2 molecules has been proven to be growth-medium dependent [52,53]. The *luxS* gene, responsible for AI-2 production, is subject to catabolic repression by glucose, thus AI-2 molecules are difficult to be detected in a growth medium containing glucose [54]. De Keersmaecher Vanderleyden [53] suggested that a final concentration of 2 mM of glucose present in the CFS could cause an inhibition to the light production in the bioassay, whereas other sugars such as galactose did not. Furthermore, the acidic conditions present in CFS could also have an impact on the bioassay, as reported in a previous study [53]. Another critical factor in the detection of AI-2 molecules is the growth stage. The best point in growth to detect AI-2 molecules in LAB is during the late exponential phase and/or the stationary phase, as the molecules are still considered to be present. This information was taken into consideration in the current study, and the supernatant was collected after 20 h of incubation at 30 °C, whereas the bacterial population was confirmed with plate counting. 

Many potentially probiotic bacteria such as *Bifidobacterium* and *Lactobacillus* having a *luxS* homologue can produce AI-2 molecules. *Lactobacillus rhamnosus* GG has been thoroughly studied regarding its ability to produce AI-2 and regulate its physiology [22], suggesting that the *luxS* gene has a central metabolic role in this strain. In a previous study [24], *Lactobacillus rhamnosus* GG and *Lactobacillus salivarius* UCC118 were found to produce AI-2 signal molecules under standard growth conditions, reaching the maximum concentration at the late exponential and stationary phase, respectively. In the same study, the AI-2 activity after an acidic shock with pH of 3 and 4 showed an increase in *Lactobacillus acidophilus* NCFM and *Lactobacillus rhamnosus* GG. Such an observation supported the hypothesis that the LuxS-mediated quorum sensing via AI-2 activity possibly plays an important role in the stress tolerance response of *Lactobacillus* species. Park et al. [51] reported various intensities of AI-2 activity in fermented kimchi products and also from the LAB obtained from these products. Strains of *Lactobacillus plantarum*, *Lactobacillus brevis*, *Lactobacillus fermentum* and *Lactobacillus garlicum* exhibited significant AI-2 activity, which was considered by the authors as an interesting characteristic for the future of fermented foods [51]. Although QS is generally reported in LAB participating in food fermentations [55], it is possible that the most dominant QS system is that of autoinducing peptides (AIP), especially for the case of fermented vegetables such as table olives [56,57,58,59].

### 3.4. Detection of Extracellular GABA

Screening of LAB for their ability to produce GABA is important for the food industry, as GABA-producing strains could be utilised as starters or adjunct cultures in fermented foods, developing GABA-enriched functional products. GABA is synthesized and exported by the glutamic acid decarboxylase (GAD) system, which is a very potent acid resistance mechanism. The GAD enzyme catalyses the proton-consuming decarboxylation of L-glutamate to GABA, which subsequently is exported by a glutamate/GABA antiporter that also imports another glutamate molecule to initiate another cycle of glutamate decarboxylation [39]. Furthermore, the GAD system can decarboxylate intracellular L-glutamate pools to produce intracellular GABA, which can be metabolised to succinate through the GABA shunt [60]. Several studies have indicated the presence of GAD system in lactic acid bacteria [29]. Generally, GABA synthesis in bacteria is related to enhanced resistance under acidic conditions. In the present study, 17 strains of LAB, with good probiotic attributes, were screened for their ability to produce GABA in vitro. Extracellular GABA was not detected in any of the strains tested (concentrations between 0.16–0.66 mM/mL) under the growth conditions tested, as presented in Table 3.

Measurements of extracellular GABA (GABA_e_) as means of quantification of the GAD system activity could potentially indicate the acid resistance of a specific microorganism. However, the quantification of the intracellular GABA (GABA_i_) is also important for the investigation of the GAD system [25]. It has to be noted that although the GAD system is widely distributed in LAB, the ability of LAB to produce GABA varies significantly [61]. Similar conclusions were made previously, for GABA production in *Listeria monocytogenes*, where different strains export GABA in different media and environmental conditions, suggesting that diverse activation signals present in different niches might activate the GAD system in different strains [62].

Several factors are considered to have a detrimental effect on GABA synthesis in vitro, such as the incubation temperature, the incubation time, and the glutamate concentrations [61,63]. The optimum temperature range for GABA synthesis is 30–37 °C, whereas at 45 °C or more, GABA is not detected, possibly due to difficulties in bacterial growth. For the detection of the highest GABA concentrations, the optimum incubation time is 48 h, whereas extra incubation time does not result in an increase in GABA values [63]. In our study, the bacterial counts of the examined strains clearly showed that the presence of L-glutamic acid did not affect their growth (9.22–9.66 log CFU/mL) (Table 3).

In a study of Yunes et al. [64], 135 human-derived strains of lactobacilli and bifidobacteria were isolated and screened for their potential to synthesize GABA. A total of 43% of the isolates were determined as GABA-producers, with the strains assigned to *Lactobacillus plantarum*, *Lactobacillus brevis*, *Bifidobacterium adolescentis*, *Bifidobacterium angulatum*, and *Bifidobacterium dentium*. On the contrary, Barrett et al. [65] reported that from the 91 human-derived lactobacilli and bifidobacteria, only 4% were GABA producers, including *Lactobacillus brevis*, *B. dentium*, *Bifidobacterium infantis*, and *B. adolescentis*, with *Lactobacillus brevis* being the one with the highest conversion ability of monosodium glutamate to GABA. Furthermore, isolates from various foods such as Italian cheeses [29], selected dairy products [66], artisanal Zlatar Cheese [67], or Nostrano cheeses made from raw alpine milk [28] were positive in GABA production by 13.86%, 50%, 28%, or 70%, respectively. The work on Italian cheeses has shown that gorgonzola and pecorino harboured a high number of GABA-producing LAB. In addition, the type of milk used for cheese manufacturing, together with the ripening period, had an impact on the GABA concentrations.

Interestingly, in our work we found no isolate producing any GABA. This is in contrast with the situation in isolates from human colon and dairy products. This might be related to the fact that LAB strains isolated from dairy products might have a higher potential in GABA production compared to strains isolated from non-dairy products [68]. Cheese possesses specific characteristics that favour the natural presence of GABA-producing LAB. The high content of L-glutamate (17.5% of the total amino acid content) in milk caseins is metabolized from LAB during the ripening process [69]. In addition to strain variability, GABA production is affected by many other factors such as the temperature, the pH, the medium composition, and other environmental factors [29,70,71]. Work by Villegas et al. [63] has shown that 3 out of 19 strains (15%) originating from amaranth and quinoa showed GABA producing ability that was low but still significantly higher than the absence of GABA producers found in this study. It is well known that the amount of GABA available in fruits and vegetables is relatively low compared to other sources such as dairy and meat products [63,72]. In a study of Karatzas et al. [39], it was reported that *L. monocytogenes*, although producing GABA in rich media, is unable to export GABA in a defined medium supplemented with L-glutamate. This suggests that additional factors present in a nutrient-rich environment play an important role in the function of the GAD system.

## 4. Conclusions

A total of 17 strains were chosen as good candidates for potentially probiotic applications, as adjunct or co-starter cultures, although more tests are required to further assess their potential as probiotics. AI-2 signal molecules and extracellular GABA were not detected under the in vitro growth conditions tested; however, further research is needed to better understand the systems involved and the mechanisms triggering their production in LAB. The lack of GABA-producing strains isolated from natural black olives seems unique among various fermented foods, although the functionality of the GAD system or the genes involved have not yet been studied.

## Figures and Tables

**Figure 1 foods-08-00640-f001:**
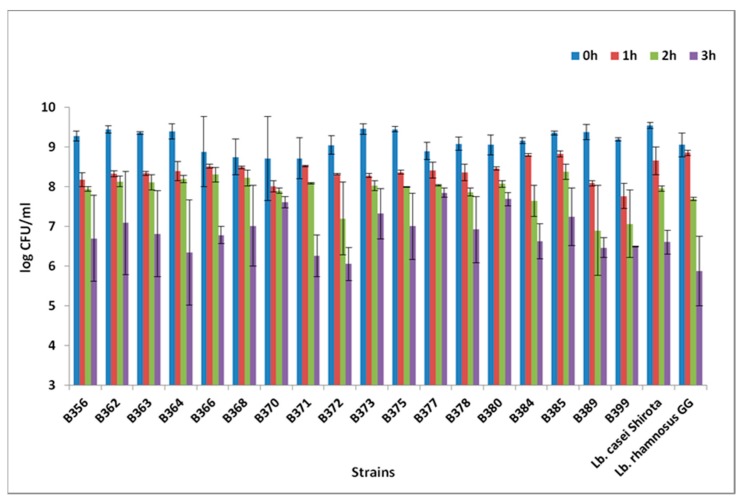
Resistance to low pH after 0, 1, 2, and 3 h of the selected strains *Lactobacillus pentosus* B356, B362, B363, B364, B366, B368, B370, B371, B377, B378, B385, B399; *Lactobacillus plantarum* B372, B373, B375, B380, B384; *Pediococcus ethanolidurans* B389; and the reference strains *Lactobacillus casei* Shirota and *Lactobacillus rhamnosus* GG.

**Figure 2 foods-08-00640-f002:**
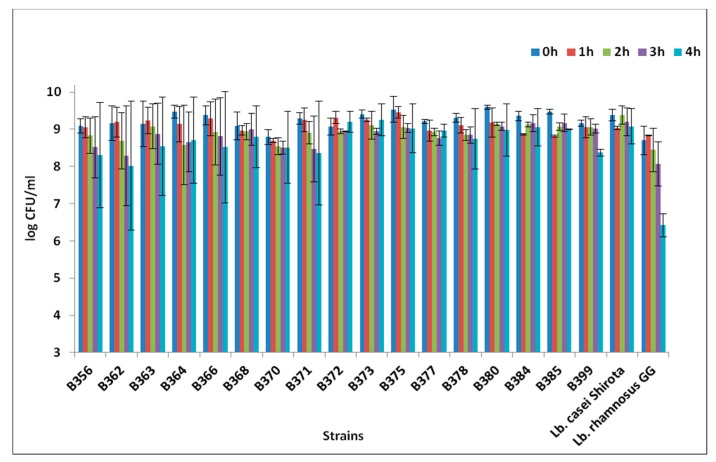
Resistance to bile salts after 0, 1, 2, 3, and 4 h of the selected strains *Lactobacillus pentosus* B356, B362, B363, B364, B366, B368, B370, B371, B377, B378, B385, B399; *Lactobacillus plantarum* B372, B373, B375, B380, B384; and the reference strains *Lactobacillus casei* Shirota and *Lactobacillus rhamnosus* GG.

**Table 1 foods-08-00640-t001:** List of lactic acid bacteria (LAB) strains used in the study and their final viable counts after exposure to pH 2.5 for 3 h.

Species	Strain	Final Counts (log CFU/mL)
*Lactobacillus plantarum*	B355	<1
*Lactobacillus plantarum*	B359	2.57 ± 1.14
*Lactobacillus plantarum*	B372	6.05 ± 0.41
*Lactobacillus plantarum*	B373	7.31 ± 0.63
*Lactobacillus plantarum*	B374	4.15 ± 0.15
*Lactobacillus plantarum*	B375	7.00 ± 0.83
*Lactobacillus plantarum*	B380	7.68 ± 0.16
*Lactobacillus plantarum*	B384	6.62 ± 0.43
*Lactobacillus pentosus*	B356	6.69 ± 1.08
*Lactobacillus pentosus*	B357	<1
*Lactobacillus pentosus*	B360	1.79 ± 1.18
*Lactobacillus pentosus*	B361	3.96 ± 0.73
*Lactobacillus pentosus*	B362	7.08 ± 1.29
*Lactobacillus pentosus*	B363	6.81 ± 1.07
*Lactobacillus pentosus*	B364	6.34 ± 1.32
*Lactobacillus pentosus*	B366	6.78 ± 0.22
*Lactobacillus pentosus*	B368	7.01 ± 1.01
*Lactobacillus pentosus*	B369	5.48 ± 0.07
*Lactobacillus pentosus*	B370	7.61 ± 0.14
*Lactobacillus pentosus*	B371	6.26 ± 0.52
*Lactobacillus pentosus*	B377	7.85 ± 0.11
*Lactobacillus pentosus*	B378	6.92 ± 0.83
*Lactobacillus pentosus*	B383	4.61 ± 0.31
*Lactobacillus pentosus*	B385	7.24 ± 0.73
*Lactobacillus pentosus*	B399	6.48 ± 0.01
*Lactobacillus pentosus*	B400	2.07 ± 0.32
*Lactobacillus pentosus*	B401	<1
*Lactobacillus pentosus*	B402	<1
*Lactobacillus paraplantarum*	B365	3.22 ± 1.56
*Pediococcus ethanolidurans*	B389	6.46 ± 0.25
*Pediococcus ethanolidurans*	B397	<1
*Lactobacillus coryniformis*	B395	2.57 ± 0.11
*Lactobacillus coryniformis*	B403	<1

**Table 2 foods-08-00640-t002:** Selected strains with probiotic potential according to in vitro tests.

Strains	Test				
	Low pH (SR%) ^A^	Bile Salts (SR%) ^B^	Bile Salts Hydrolase ^C^	Haemolytic Activity	Antimicrobial Activity
*Lactobacillus pentosus* B356	72.17	91.38	1	γ	-
*Lactobacillus pentosus* B362	75.05	87.47	1	γ	-
*Lactobacillus pentosus* B363	72.84	93.42	1	γ	-
*Lactobacillus pentosus* B364	67.55	91.88	1	γ	-
*Lactobacillus pentosus* B366	76.38	90.83	1	γ	-
*Lactobacillus pentosus* B368	80.16	96.78	1	γ	-
*Lactobacillus pentosus* B370	87.38	96.83	1	γ	-
*Lactobacillus pentosus* B371	71.86	90.04	1	γ	-
*Lactobacillus plantarum* B372	66.93	101.29	1	γ	-
*Lactobacillus plantarum* B373	77.40	98.46	1	γ	-
*Lactobacillus plantarum* B375	74.12	94.60	1	γ	-
*Lactobacillus pentosus* B377	88.19	97.26	1	γ	-
*Lactobacillus pentosus* B378	76.20	93.88	0	γ	-
*Lactobacillus plantarum* B380	84.88	93.53	1	γ	-
*Lactobacillus plantarum* B384	72.35	96.76	1	γ	-
*Lactobacillus pentosus* B385	77.41	95.03	1	γ	-
*Lactobacillus pentosus* B399	70.56	91.32	0	γ	-

^A^ Survival rate after 3 h in low pH. ^B^ Survival rate after 4 h in bile salts. ^C^ 0: no hydrolysis; 1: partial hydrolysis.

**Table 3 foods-08-00640-t003:** Relative autoinducer-2 (AI-2) activity and GABA values of the selected LAB exhibiting probiotic potential.

Strains	AI-2 Activity			GABA Determination	
	Bacterial Counts ^A^	Relative AI-2 Activity ^B^	Bacterial Counts ^C^	pH ^D^	GABA ^E^
*Lactobacillus pentosus* B356	7.57 ± 0.15	0.68 ± 0.22 ^a^	9.52 ± 0.17	3.92 ± 0.00	0.30 ± 0.05 ^a^
*Lactobacillus pentosus* B362	7.55 ± 0.06	0.57 ± 0.15 ^a^	9.40 ± 0.17	3.91 ± 0.00	0.16 ± 0.03 ^a^
*Lactobacillus pentosus* B363	7.13 ± 0.22	0.70 ± 0.27 ^a^	9.52 ± 0.17	4.09 ± 0.01	0.52 ± 0.49 ^a^
*Lactobacillus pentosus* B364	7.26 ± 0.08	0.58 ± 0.28 ^a^	9.44 ± 0.17	4.00 ± 0.04	0.40 ± 0.11 ^a^
*Lactobacillus pentosus* B366	7.47 ± 0.09	0.48 ± 0.11 ^a^	9.66 ± 0.42	3.92 ± 0.00	0.30 ± 0.07 ^a^
*Lactobacillus pentosus* B368	7.51 ± 0.37	0.50 ± 0.16 ^a^	9.32 ± 0.06	4.02 ± 0.03	0.27 ± 0.06 ^a^
*Lactobacillus pentosus* B370	8.30 ± 0.24	0.56 ± 0.17 ^a^	9.49 ± 0.15	4.00 ± 0.04	0.39 ± 0.06 ^a^
*Lactobacillus pentosus* B371	7.73 ± 0.15	0.58 ± 0.24 ^a^	9.48 ± 0.19	3.93 ± 0.00	0.46 ± 0.08 ^a^
*Lactobacillus plantarum* B372	7.28 ± 0.14	1.39 ± 0.45 ^b^	9.51 ± 0.07	4.05 ± 0.03	0.66 ± 0.05 ^a^
*Lactobacillus plantarum* B373	8.11 ± 0.52	1.28 ± 0.33 ^b^	9.31 ± 0.01	3.95 ± 0.03	0.56 ± 0.51 ^a^
*Lactobacillus plantarum* B375	8.15 ± 0.22	1.34 ± 0.68 ^b^	9.60 ± 0.43	3.93 ± 0.00	0.53 ± 0.13 ^a^
*Lactobacillus pentosus* B377	6.98 ± 0.13	0.53 ± 0.27 ^a^	9.28 ± 0.24	3.99 ± 0.06	0.30 ± 0.07 ^a^
*Lactobacillus pentosus* B378	7.18 ± 0.50	0.32 ± 0.12 ^a^	9.41 ± 0.42	3.92 ± 0.00	0.14 ± 0.10 ^a^
*Lactobacillus plantarum* B380	7.34 ± 0.01	1.57 ± 1.16 ^b^	9.29 ± 0.16	3.92 ± 0.00	0.40 ± 0.05 ^a^
*Lactobacillus plantarum* B384	6.42 ± 0.16	0.38 ± 0.14 ^a^	9.27 ± 0.27	4.06 ± 0.01	0.42 ± 0.17 ^a^
*Lactobacillus pentosus* B385	7.15 ± 0.22	0.54 ± 0.15 ^a^	9.22 ± 0.32	3.95 ± 0.04	0.34 ± 0.21 ^a^
*Lactobacillus pentosus* B399	6.94 ± 0.15	0.48 ± 0.19 ^a^	9.49 ± 0.02	3.95 ± 0.03	0.23 ± 0.02 ^a^

^A^ Bacterial counts (log CFU/mL) after 20 h in ¼ strength brain heart infusion (BHI) at 30 °C are presented as mean ± standard deviation. ^B^ Relative AI-2 activity was calculated as the ratio of the luminescence of the test sample (CFS_LAB_) to that of the control (negative) and is presented as mean ± standard deviation. Values with different letters are significantly different (*p* < 0.05). ^C^ Bacterial counts (log CFU/mL) after 48 h in MRS broth supplemented with L-glutamic acid at 37 °C are presented as mean ± standard deviation. ^D^ pH after 48 h in MRS broth supplemented with L-glutamic acid at 37 °C is presented as mean ± standard deviation. ^E^ γ-Aminobutyric acid (GABA) values (mM) are presented as mean ± standard deviation. Values with different letters are significantly different (*p* < 0.05).

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
