# Peer review of "Ιn Vitro Screening of γ-Aminobutyric Acid and Autoinducer-2 Signalling in Lactic Acid Bacteria Exhibiting Probiotic Potential Isolated from Natural Black Conservolea Olives"

_foods, 2019, doi:10.3390/foods8120640_

Round 1

Reviewer 1 Report

The manuscript is very interesting, and the topic of research is current. The aim of the work is clearly defined. The scope of research is clearly described. Despite this, the manuscript needs to be completed:

2.1. In vitro tests simulating GI tract conditions:

There is no justification in the research methodology why pH 2.5 and 8.0 were chosen. After all, the pH values are not constant in the digestive system.

2.3. Screening for AI-2 activity & 2.4. Detection of extracellular GABA:

The research methodology is not complete, it is not fully described (for example, there are no significant measurement parameters).

Author Response

Response to Comments from Reviewer 1

Reviewer 1 comments:

The manuscript is very interesting, and the topic of research is current. The aim of the work is clearly defined. The scope of research is clearly described. Despite this, the manuscript needs to be completed: 2.1. In vitro tests simulating GI tract conditions: There is no justification in the research methodology why pH 2.5 and 8.0 were chosen. After all, the pH values are not constant in the digestive system.

Response:

It is common to use an acidic challenge at pH 2.5 as an indication of survival in the stomach. We do not see any problem with these experiments as they are common practice in the area and also in our research group. The pH of bile as produced by the liver is close to 8. When it is secreted in the gallbladder it has a pH of 7.5. The pH in the duodenum can be lower than that, however, the pH is not really stable as acid and bile are secreted in there. It is common practice to do experiments in bile without acidifying the solution since bile salts precipitate and acidic conditions making difficult the interpretation and the consistency of results. It is understandable that the exact conditions in the GI tract are not simulated exactly since they are themselves really variable as well. In addition, these compromises are completely acceptable by the scientific community as we need to challenge the cells with one stress at a time to better understand mechanisms and microbial behaviour. Therefore, if in the bile experiments we had lowered the pH we would have a combined acidic and bile stress applied simultaneously making interpretations very difficult.

Furthemore, the pH of the stomach can be higher than 2.5, when food is consumed, and can reach values up to 4-5 in such cases. However, most in vitro assays have been developed to select strains that withstand extreme low pH values, giving information on their resistance in the worst case scenario that can be encountered in the stomach (pH 2.5). The pH value (2.5) used in this study is very selective and even though it is not constant in the digestive system as pointed by the reviewer, it assures the isolation of the very acid-tolerant strains as reported elsewhere (Pennacchia et al., 2004, Selection of Lactobacillus strains from fermented sausages for their potential use as probiotics, Meat Science, 67, 309-317).    

2.3. Screening for AI-2 activity & 2.4. Detection of extracellular GABA: The research methodology is not complete, it is not fully described (for example, there are no significant measurement parameters).

Response:

 Regarding the AI-2 activity, the methodology used is a standard one and widely used by the scientific community, and the results were expressed as Relative AI-2 activity. Also other parameters that could have affected the findings, such as the bacterial counts were also recorded and given in Table 3 for each tested strain. Regarding the GABA measurement, we have overnight cultures of the strains, where we do not adjust the pH to acidic values since the LAB isolates have reduced the pH of their cultures to levels between 3.91 and 4.09. These values are within the optimal pH where the GAD system of these bacteria is active and would produce GABA without any pH adjustment. Therefore, the fact that we do not detect any GABA confirms that the strains do not produce GABA. In the literature, GABA production of lactic acid bacteria is not performed with pH adjusted and overnight cultures are used. The only other parameter we had not mentioned was the concentration of L-glutamate with which, we supplemented MRS. This was 10 mM L-glutamate and has now been added in the text.

Reviewer 2 Report

The manuscript aimed at investigating probiotic potential of a group of lactic acid bacteria in vitro. Here authors claim to have used “simulated gut condition”, only pH and bile salts (especially separately) don’t make up for actual gut. Did the authors consider anaerobic or microaerophilic condition? Authors used very high inoculum size, does the results reproduce with lower numbers (3-5 logs range)? Authors need to justify how examining 33 random LAB strains may advance scientific knowledge, to be more specific, in what aspect the study may be novel?

Author Response

Response to Comments from Reviewer 2

Reviewer 2 comments:

The manuscript aimed at investigating probiotic potential of a group of lactic acid bacteria in vitro. Here authors claim to have used “simulated gut condition”, only pH and bile salts (especially separately) don’t make up for actual gut.

Response:

We think that it is actually better that we performed each test separately. The way we have put all tests under this umbrella looks like we simulate all GI conditions. However, we just give an indication of how these bacteria would survive in different parts of the GI tract. It has to be noted that this is not a definite proof and it is impossible to simulate all parts of the GI tract. The survival at pH 2.5 is clearly an indicator of survival in the stomach and the bile tests survival in the duodenum. These tests are still considered as tests simulating GI conditions.

Did the authors consider anaerobic or microaerophilic condition?

Response:

For the cultures used in the tests simulating GI conditions and AI-2 detection, the culture growth is aerobic in the first couple of hours followed by microaerophilic-anaerobic conditions for most of the growth. For the cultures used for GABA detection, the conditions were anaerobic, as described in section 2.4.

Authors used very high inoculum size, does the results reproduce with lower numbers (3-5 logs range)?

Response:

When testing strains for probiotic potential, it makes sense to apply the stresses at a high inoculum size and this is the common practice used in probiotic research. For a strain to confer a health benefit to the host, a high amount of cells is to be administered as per definition of FAO/WHO (2002) for probiotics. As a minimum, probiotic strains need to be present in probiotic foods in levels of 108 or 109 CFU per ml or g, while these levels are expected to be lower after encountering the stresses during the passage through GIT (as a minimum the accepted levels are 106 CFU per ml or g). Therefore, testing lower inoculum size (3-5 logs), does not make sense, since by definition such amounts are not considered high enough to confer any potential health benefit to the host.

Authors need to justify how examining 33 random LAB strains may advance scientific knowledge, to be more specific, in what aspect the study may be novel?

Response:

We believe that the results obtained from the strains isolated from Conservolea table olives can give useful insights on the characteristics and physiology of lactic acid bacteria naturally found in this product and expand the scientific knowledge. Regarding the probiotic characteristics of these isolates, we can conclude that table olives is a product that can be a source of probiotic bacteria, and our results are in accordance with previous studies found in the literature. However, studies on AI-2 signal detection in situ in fermented products such as table olives are scarce, and most of the studies deal with testing individual isolates for AI-2 production in vitro. Testing of isolates from table olives for AI-2 production, is something that was missing from the literature and therefore we believe that this makes the study novel. In addition, the fact that no AI-2 activity was observed, it is an important result, which is different from what is reported in studies for fermented products. Regarding the GABA detection, the results were considered as “unexpected” since we believed that at least some of the strains would have the ability to produce GABA under the conditions tested, similarly to previous reports and observations. As a result these “unexpected” findings have challenged us for the future to further investigate the conditions under which GABA can be produced, which parameters could affect the phenomenon, which of these strains have the genes ect. For all these reasons, we strongly believe that our study can contribute to the scientific knowledge and provide new aspects in this field of study. In addition, the strains of lactic acid bacteria with the best performance could be selected as starter cultures in table olive fermentation providing functional characteristics to the final product. Thus the novelty of this work is related to the characterization of the functional traits of lactic acid bacteria that will be used as a second step in the fermentation of table olives as a source of potential probiotic strains.

Round 2

Reviewer 2 Report

Again, only pH and bile salts do not make up for actual gut. Therefore, the experimental conditions tested in this study do not simulate gut condition in any way. The authors should reword “simulated gut condition” throughout the manuscript.

Need to add info on biological and technical replication in methods.

Lines 227-229: Check for ambiguity “In the present study none of the 17 strains, previously selected for their good performance under simulated GI tract conditions, exhibited α- or β-haemolysis, while none of them exhibited γ-haemolysis (no haemolysis).”

Author Response

 Authors’ Response to the Reviewers’ Comments: We appreciate the time and efforts by the editor and referees in reviewing this manuscript. We have acknowledged, replied and included all the suggestions listed in the review report, and we believe that the revised version can meet the journal publication requirements. All the changes are highlighted with yellow color in the revised manuscript.

Response to Comments from Reviewer 1: Reviewer 1 comments:Again, only pH and bile salts do not make up for actual gut. Therefore, the experimental conditions tested in this study do not simulate gut condition in any way. The authors should reword “simulated gut condition” throughout the manuscript.

Response: We have removed the “simulated gut condition” throughout the manuscript and instead we have used “Low pH assay” and “Bile salts assay”.

 Need to add info on biological and technical replication in methods.

Response:  In every section of materials and methods details regarding replications have been added.

Lines 227-229: Check for ambiguity “In the present study none of the 17 strains, previously selected for their good performance under simulated GI tract conditions, exhibited α- or β-haemolysis, while none of them exhibited γ-haemolysis (no haemolysis).”

Response: We thank the reviewer, it was a typing mistake. The sentence now has been corrected “In the present study none of the 17 strains, previously selected for their good performance in low pH and bile assays, exhibited α- or β-haemolysis, while all of them exhibited γ-haemolysis (no haemolysis).”
